# MultiNet: Adaptive Multi-Viewed Subgraph Convolutional Networks for Graph Classification

**Xinya Qin, Lu Bai**[1,*]**, Lixin Cui**[2]**, Ming Li**[3,4]**, Hangyuan Du**[5]**, Edwin R. Hancock**[6]
[1]School of Artificial Intelligence, Beijing Normal University, Beijing, China.
[2]School of Information, Central University of Finance and Economics, Beijing, China.
[3]Zhejiang Institute of Optoelectronics, Jinhua, China.
[4]Zhejiang Key Laboratory of Intelligent Education Technology and Application,
Zhejiang Normal University, Jinhua, China.
[5]School of Computer and Information Technology, Shanxi University, Taiyuan, China.
[6]Department of Computer Science, University of York, York, United Kingdom.
XinyaQin@mail.bnu.edu.cn, bailu@bnu.edu.cn

## Abstract

The problem of over-smoothing has emerged as a fundamental issue for Graph Convolutional Networks (GCNs). While existing efforts primarily focus on enhancing the discriminability of node representations for node classification, they tend to overlook the over-smoothing at the graph level, significantly influencing the performance of graph classification. In this paper, we provide an explanation of the graph-level over-smoothing phenomenon and propose a novel Adaptive Multi-Viewed Subgraph Convolutional Network (MultiNet) to address this challenge. Specifically, the MultiNet introduces a local subgraph convolution module that adaptively divides each input graph into multiple subgraph views. Then a number of subgraph-based view-specific convolution operations are applied to constrain the extent of node information propagation over the original global graph structure, not only mitigating the over-smoothing issue but also generating more discriminative local node representations. Moreover, we develop an alignment-based readout that establishes correspondences between nodes over different graphs, thereby effectively preserving the local node-level structure information and improving the discriminative ability of the resulting graph-level representations. Theoretical analysis and empirical studies show that the MultiNet mitigates the graph-level over-smoothing and achieves excellent performance for graph classification.

## 1 Introduction

Graph data analysis has become an important research area for deep learning, and has been widely employed in various fields, including bioinformatics [36, 17], social networks [21, 22], and recommendation systems [33, 9]. However, traditional neural networks are originally designed for grid-like data, such as images or sequences. They struggle to handle graphs due to the irregular structures and non-Euclidean properties of graph data. To address these limitations, Graph Neural Networks (GNNs) have emerged as powerful tools for learning on graph-structured data.

In recent years, Graph Convolutional Networks (GCNs) have become the most popular architecture in GNN research. The early spectral-based GCNs, such as the Spectral GNN [7], ChebNet [13], and Vanilla GCN [18], interpret graphs as signals and define convolution operations in the spectral domain. However, these methods are computationally expensive and rely on the eigenvalue spectrum of the graph Laplacian matrix, making them less scalable. To improve the efficiency, an alternative

---

*The Corresponding Author

39th Conference on Neural Information Processing Systems (NeurIPS 2025).

line of spatial-based GCNs has been introduced recently, such as the GraphSAGE [16] and GIN [34] that perform convolutions by aggregating the information from neighboring nodes without spectral decomposition. While these spatial GCN approaches are usually computationally efficient, they still suffer from several challenges.

One notable challenge for most existing GCNs is the over-smoothing phenomenon [20, 24, 8], where an increasing number of layers leads to excessive propagation of node features across the global graph structure. Since the information diffuses through multiple hops, the initially discriminative local node-level structural features tend to lose their discriminative power, resulting in increasingly similar node representations. This homogenization weakens the ability of GCNs to capture local node differences and ultimately degrades the performance on node classification. To overcome the shortcoming, a variety of mitigation strategies have been developed, including the normalization [25], regularization [40], dynamic connections [26, 27], and residual connections [19, 10].

Nevertheless, most existing studies mainly investigate the convergence of node representations within a single graph structure. The impact of over-smoothing at the graph level has not been sufficiently explored by now. In this paper, we argue that deep GCNs also tend to generate extremely similar representations for different graphs, leading to the graph-level over-smoothing and ultimately influencing the performance of graph classification. This issue is due to the following two key factors. First, when the node information propagates over multiple layers, the resulting node representations tend to be increasingly similar, leading to a significant information loss of local distinct nodes. For graphs with similar node information distributions, this will cause the node representations over all graphs to converge. Second, the unordered nature of graph nodes forces GNNs to employ permutation-invariant global readout functions, such as summation, mean, or max pooling, for generating graph-level representations. However, the simplicity of these operations treats all nodes equally, thereby further ignoring the distinct structural information of local nodes associated with similar node representations. As a result, the graphs with structurally distinct local-level node features may still produce similar graph-level representations, thereby decreasing the discriminative power of graph classification models.

To indicate how the graph-level over-smoothing phenomenon influences the performance of GCNs, Figure 1 reports the graph classification accuracy and the average cosine distance between graph representations (defined in Section 5.3) with varying network depths, associated with three popular GCN models. The results reveal that both the classification accuracies and the representation diversities simultaneously degrade with the deeper network layers (i.e., more than eight layers), and indicate that the models struggle to distinguish between different graph structures. This observation provides empirical evidence that the graph-level over-smoothing arising in GCNs has a negative influence on graph classification.

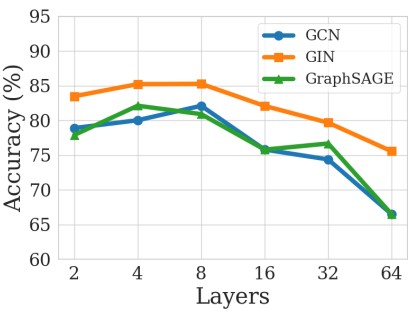
(a) The graph classification accuracy.

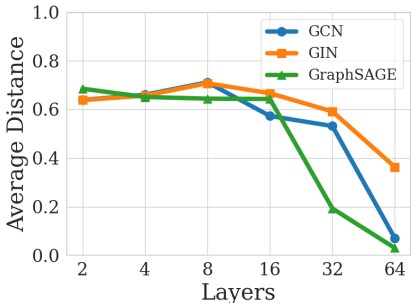
(b) The average distance.

Figure 1: The impact of increasing layers on the MUTAG dataset.

The aim of this paper is to develop a novel **Adaptive Multi-Viewed Subgraph Convolutional Network (MultiNet)**, that can mitigate the influence of the graph-level over-smoothing problem for graph classification tasks. The key innovation lies in a local subgraph convolution module as well as an alignment-based readout mechanism. The former can restrict the excessive message propagation, preserving local distinct node features and alleviating the feature homogenization. The latter maintains the local information and structural information for the resulting graph-level representations, enhancing the ability to distinguish different graphs. The main contributions of this work are summarized as follows:

- We propose a local subgraph convolution module that can adaptively divide the original input graph into multiple structurally distinct subgraph views. Within each subgraph view, we define an attention-based convolution operation to capture the discriminative local node features. Then we introduce a feature fusion strategy that integrates the complementary information from different subgraph perspectives. This design can effectively preserve the local distinct node information and alleviate the node-level over-smoothing within each individual graph.

- We develop an alignment-based readout mechanism to overcome the limitations of traditional global readout functions in preserving local features and structural information. By establishing consistent cluster-level correspondences across different graphs, our method enables the use of a more expressive readout function during graph aggregation, thereby enhancing the discriminative capability of the learned graph representations. This mechanism effectively preserves local features and structural distinctions extracted from the local subgraph convolution modules within the graph-level representation, thus alleviating the graph-level over-smoothing problem.

- We provide both theoretical and empirical analyses, and demonstrate that the proposed MultiNet can effectively alleviate the graph-level over-smoothing. Experiments on benchmark datasets indicate that the MultiNet can outperform state-of-the-art methods on graph classification tasks.

This paper is organized as follows. Section 2 discusses the node-level and the graph-level over-smoothing problem. Section 3 details the MultiNet model. Section 4 analyzes the mitigation effect of the MultiNet. Section 5 presents experiments and results. Section 6 concludes this work.

## 2   The Over-smoothing Problem for GCNs

### 2.1   The Node-level Over-Smoothing Phenomenon

The over-smoothing phenomenon is a fundamental challenge in GCNs, where repeated message passing causes node representations to become indistinguishable, ultimately impairing performance on node-level tasks and potentially affecting graph-level tasks. Various works provide different perspectives on the underlying mechanisms of the node-level over-smoothing. For example, [20] models graph convolutional layers as a form of Laplacian smoothing, showing that deep networks drive the node representations toward a space where they become indistinguishable. From a spectral viewpoint, [24] interprets each graph convolution layer as a low-pass filter that attenuates the high-frequency components of the graph signal. While effective for noise reduction, excessive filtering blurs discriminative information, thereby hindering effective classification.

To address this issue, various strategies have been introduced. DropEdge [25] applies stochastic edge removal as a form of implicit regularization. PairNorm [40] stabilizes feature diversity by preserving pairwise distances between node embeddings. GraphCON [26] reformulates message passing as a nonlinear dynamical system to enhance representation stability. $G^2$ [27] introduces a gradient-based gating mechanism that adaptively halts message propagation per node. Residual-based methods like Res-GCN[19] and GCNII [10] incorporate skip connections to retain initial features throughout the network. However, these approaches primarily target the node-level representations and do not explicitly restrict information diffusion. In this work, we revisit the over-smoothing problem from a graph-level perspective and present a method that effectively alleviates this phenomenon.

### 2.2   The Graph-Level Over-smoothing Phenomenon

While prior studies have primarily focused on the node-level over-smoothing problem, the graph-level over-smoothing phenomenon remains underexplored. Although [31] introduces a strategy to alleviate over-smoothing and applies it to graph classification tasks, it still primarily targets the node-level issue and lacks experiments that characterize the graph-level over-smoothing. We argue that excessive convolution across same-domain graphs can lead to indistinguishable graph representations, where the representations of different graphs become excessively similar after the readout stage, making it difficult for the model to differentiate between graphs and thereby impairing graph classification performance. Here we begin with a formal definition, followed by theoretical analysis.

**Definition 1 (The Graph-Level Over-Smoothing)** Let $\mathcal{G} = \{G_1, G_2, \ldots, G_N\}$ denote a set of undirected and connected graphs, and let $H \in \mathbb{R}^{N \times d}$ be the graph representation matrix after readout

operation and $H_i \in \mathbb{R}^d$ be the graph representation of graph $G_i$. Define a similarity measure $\mu : \mathbb{R}^{N \times d} \to \mathbb{R}_{\geq 0}$ satisfying the following properties:

- a) $\exists c \in \mathbb{R}^d$ s.t. $H_i = c$ for all graphs i $\in \{1, \cdots N\} \Leftrightarrow \mu(H) = 0$, for $H \in \mathbb{R}^{N \times d}$;
- b) $\mu(H + Z) \leq \mu(H) + \mu(Z)$, for all $H, Z \in \mathbb{R}^{N \times d}$.

If the representations of different graphs after multiple graph convolution layers tend to become indistinguishable, i.e., $\mu(H) \to 0$, we say that **the graph-level over-smoothing** has occurred.

**Assumption 1 (The Node Feature Distribution Consistency)** Given any two graphs $G_i, G_j \in \mathcal{G}$ with initial node feature matrices $X_i$ and $X_j$, we assume that their feature distributions are statistically consistent, especially in datasets drawn from a shared domain or labeling space. Specifically, the average node feature vectors $x_i, x_j$ of $X_i$ and $X_j$ satisfy $x_i \approx x_j$, indicating that the node features across graphs follow similar distributions.

In graph classification datasets, samples typically originate from similar domains, and node features are represented as sparse one-hot encodings. Hence, it is reasonable to assume that the distributions of node features across different graphs are largely consistent. In subsection 5.3 and Table 2, we provide empirical evidence showing that node features in graph classification datasets from a specific domain exhibit distributional consistency, thereby supporting the validity of Assumption 1.

**Theorem 1** Under Assumption 1, excessive and unconstrained message passing, together with overly simple readout functions, tends to lead to the graph-level over-smoothing.

**Proof:** According to the analysis in [20], the spectral radius of the normalized adjacency matrix $\tilde{A}_i$ is 1, and its eigenvalues $\{\lambda_k\}_{k=1}^{n_i}$ satisfy $\lambda_k \in (-1, 1]$ [18], where $n_i$ is the number of nodes in graph $G_i$. Therefore, as the number of layers $L$ increases, the terms corresponding to eigenvalues with $|\lambda_k| < 1$ decay exponentially. As $L \to \infty$, only the spectral component corresponding to the dominant eigenvalue $\lambda_k = 1$ remains, leading to the over-smoothing problem. In this case, considering the simplest form of a graph convolution layer, where the activation functions are linear and the weight matrix is assumed to be the identity matrix, the node representations $X_i^{(L)}$ after $L$ layer convolutions will eventually converge to the direction related to the average feature vectors, i.e.,

$$\lim_{L \to \infty} X_i^{(L)} = \mathbf{1}_{n_i} x_i^\top, \tag{1}$$

where $\mathbf{1}_{n_i}$ is the all-ones vector. This convergence indicates that the feature differences between nodes are smoothed out, leading to the loss of local-level information. Moreover, common readout functions (such as summation, averaging, or maximizing) treat all nodes in the graph equally, losing the ability to capture the structural information of the graph. For instance, consider the averaging readout used to obtain the graph-level representations:

$$H_i = \frac{1}{n_i} \sum_{k=1}^{n_i} X_{ik}^{(L)}, \tag{2}$$

where $k$ is the node index. Since node features tend to converge, using such a global readout function is equivalent to directly averaging the original features, rendering the convolution operation ineffective for distinguishing different graphs. According to Assumption 1, $x_i \approx x_j$, we have $H_i \approx H_j$, which leads to $\mu(H) \to 0$. Therefore, the graph representations tend to become indistinguishable, and the graph-level over-smoothing has occurred. $\square$

## 3 MultiNet: The Adaptive Multi-Viewed Subgraph Convolutional Network

### 3.1 The Overall Framework

In this paper, we propose a novel MultiNet to alleviate the graph-level over-smoothing problem. As shown in Figure 2, the overall framework of the MultiNet consists of three main components: the Global Graph Convolution, the Local Graph Convolution, and the Alignment-Based Readout. First, the **Global Graph Convolution** layer performs initial feature abstraction. Then, the **Local Subgraph Convolution Module** divides the graph into multiple subgraph views and applies adaptive subgraph convolutions to extract discriminative local features, followed by integrating information across views. Finally, the **Alignment-Based Readout** module aligns nodes across different graphs to generate a more discriminative graph-level representation.

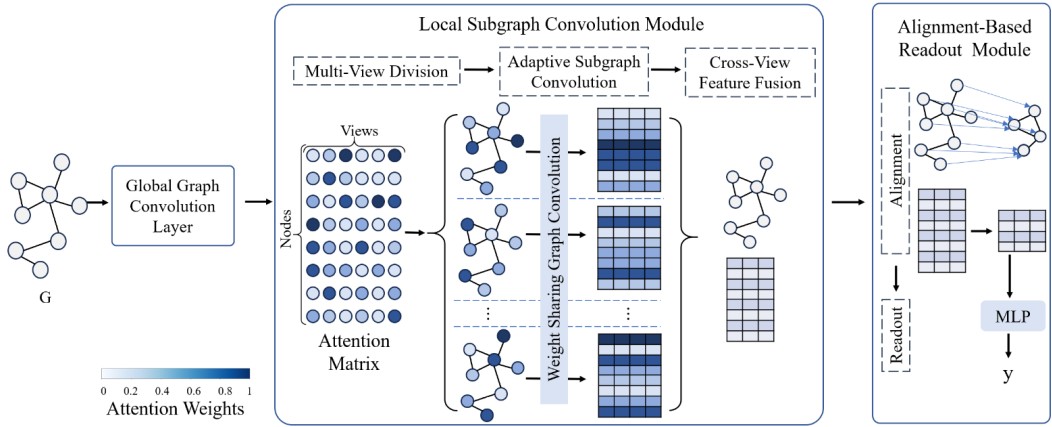

Figure 2: Overall framework of the proposed MultiNet.

## 3.2 The Global Graph Convolution Operation

Let $G(A, X)$ denote the input sample graph with $n$ nodes, where $A \in \mathbb{R}^{n \times n}$ is the adjacency matrix and $X \in \mathbb{R}^{n \times d}$ is the node feature matrix. For attributed graphs, the node features are initialized using one-hot encodings of node labels. For unattributed graphs, we use one-hot encodings of node degrees instead. In our proposed MultiNet, we begin by applying a global graph convolution layer to the input graph $G(A, X)$ to perform initial feature extraction, producing the initial node features. For notational convenience, we denote the output of the global graph convolution as $X^{(0)} \in \mathbb{R}^{n \times d^{(0)}}$. Here we employ the basic convolution operation of the Vanilla GCN defined by [18], i.e.,

$$X^{(0)} = \sigma \left( \tilde{A} X W_G \right). \tag{3}$$

Here $W_G \in \mathbb{R}^{d \times d^{(0)}}$ is the learnable weight matrix of the graph convolution layer, and $\sigma(\cdot)$ is a non-linear activation function (e.g., ReLU). The normalized adjacency matrix $\tilde{A}$ is computed as $\tilde{A} = \hat{D}^{-1/2} \hat{A} \hat{D}^{-1/2}$, where $\hat{A} = A + I$ includes self-loops and $\hat{D}$ is the degree matrix of $\hat{A}$.

## 3.3 The Local Subgraph Convolution Module

We define a novel local subgraph convolution module to adaptively capture local structural patterns from multiple subgraph views. Specifically, the proposed local subgraph convolution module consists of three key components, i.e., the Multi-View Division, the Adaptive Subgraph Convolution, and the Cross-View Feature Fusion.

### 3.3.1 The Multi-View Subgraph Division

We propose a multi-view division strategy that adaptively assigns the graph into multiple local subgraphs. Specifically, we use a graph convolution layer to learn an attention weight matrix $P \in \mathbb{R}^{n \times m}$, where each element $P_{ij}$ represents the attention weight that softly assigns node $i$ of the original graph to the $j$-th subgraph view. A larger value of $P_{ij}$ implies that node $i$ is more important in the $j$-th subgraph view. This process enables diverse local substructure modeling and emphasizes the relative importance of nodes across different perspectives. Formally,

$$P = \text{softmax}(\tilde{A} X^{(0)} W_P), \tag{4}$$

where $W_P \in \mathbb{R}^{d^{(0)} \times m}$ is a learnable weight matrix for view assignment. The row-wise softmax function ensures that the attention weights across all views for each node sum to 1.

In the $l$-th layer of subgraph convolution, we maintain a three-dimensional node representation tensor $\mathcal{H}^{(l)} \in \mathbb{R}^{n \times m \times d^{(l)}}$. We denote the slice corresponding to view $j$ by $H_j^{(l)} = \mathcal{H}_{:,j,:}^{(l)} \in \mathbb{R}^{n \times d^{(l)}}$, so that the tensor can be written compactly as $\mathcal{H}^{(l)} = [H_1^{(l)}, \ldots, H_m^{(l)}]$. To initialize the view-specific

feature matrices, we weight the original node features using the learned attention matrix:

$$H_j^{(0)} = (\text{diag}(P_j)) X^{(0)}, \qquad j = 1, \dots, m, \tag{5}$$

where $P_j \in \mathbb{R}^n$ denotes the $j$-th column of the assignment matrix $P$, and $\text{diag}(P_j)$ is the diagonal matrix of $P_j$.

### 3.3.2 The Adaptive Subgraph Convolution Operation

Unlike the standard aggregation scheme of GCNs, we adopt a constrained message-passing strategy, where the information propagation of each node is regulated by its attention score. While this operation shares similarities with the Graph Attention Network (GAT, [32]), the attention scores here are learned globally, reflecting the overall importance of each node within this view, rather than computed from local neighborhoods. The updated representation is computed as

$$H_j^{(l+1)} = \sigma\big(\tilde{A}\,(\text{diag}(P_j))\,H_j^{(l)}\,W^{(l)}\big), \tag{6}$$

where $W^{(l)} \in \mathbb{R}^{d^{(l)} \times d^{(l+1)}}$ denotes the convolution weight at the $l$-th layer. The convolution weight is shared across subgraph views. Here, we use the normalized adjacency of the original graph, $\tilde{A}$, and the view-specific effect is realized via the node-wise gating matrix $\text{diag}(P_j)$.

### 3.3.3 The Cross-View Feature Fusion

After $L$ convolutions, we fuse features from all subgraph views to obtain comprehensive node representations. Specifically, we concatenate all view-specific node features along the feature dimension, resulting in a matrix of size $n \times md^{(L)}$, which is then passed through a Multi-Layer Perceptron (MLP) to produce a unified node feature matrix $Z \in \mathbb{R}^{n \times d'}$, i.e.,

$$Z = \text{MLP}([H_1^{(L)} \,\|\, H_2^{(L)} \,\|\, \cdots \,\|\, H_m^{(L)}]). \tag{7}$$

### 3.4 The Alignment-based Readout

Inspired by some works that study node alignment mechanisms in graphs [2, 1, 12], we propose a structure-aware graph readout mechanism based on alignment. This mechanism helps avoid the limitations of simple global readout functions, which tend to neglect the distinct structural information and local differences between nodes, thus alleviating the graph-level over-smoothing phenomenon. By establishing a reliable ordered correspondence for node clusters, this mechanism leverages a more expressive MLP for graph-level readout.

Specifically, we first assign nodes to $s$ clusters. To ensure that clusters are consistently aligned across different graph samples, we learn a shared cluster assignment function parameterized by $W_S \in \mathbb{R}^{d' \times s}$, which generates an assignment probability matrix $S \in \mathbb{R}^{n \times s}$ for each graph through a graph convolution layer, i.e.,

$$S = softmax(\tilde{A}\,Z\,W_S). \tag{8}$$

Then, we use this matrix to perform alignment for node features, i.e.,

$$\tilde{Z} = S^\top Z. \tag{9}$$

Since the aligned node features $\tilde{Z} \in \mathbb{R}^{s \times d'}$ are now order-consistent across samples, we first flatten $\tilde{Z}$ into a vector of dimension $s\,d'$ and then apply a Multi-Layer MLP to assign different importance weights to the aligned clusters, producing the graph-level representation $y$, i.e.,

$$y = \text{MLP}(\text{flatten}(\tilde{Z})). \tag{10}$$

## 4 Theoretical Analysis: How the MultiNet Works?

In this section, we theoretically analyze why the proposed MultiNet effectively mitigates the graph-level over-smoothing phenomenon. The proposed MultiNet mitigates the graph-level over-smoothing phenomenon from two aspects: First, the local subgraph convolution module limits the extent of node information propagation, reducing the loss of local node-level information and alleviating the

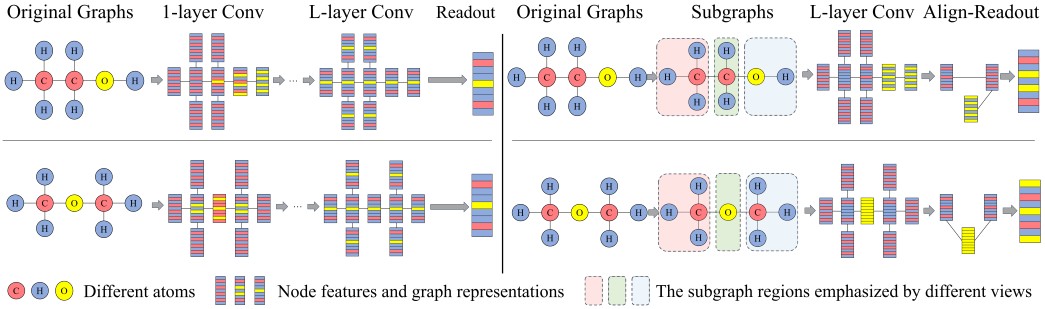

Figure 3: Toy examples to compare the conventional GCNs (left) and our MultiNet (right).

node-level over-smoothing problem. Second, the structure-aware readout mechanism aligns nodes across different graphs and employs a more expressive MLP for graph readout, thereby reducing the loss of structural information and mitigating the graph-level over-smoothing phenomenon.

Specifically, in the local subgraph convolution module, the propagation kernel in the $j$-th view defined in Equation 6 can be equivalently expressed as:

$$\tilde{A}_j = \tilde{A} \left( \mathrm{diag}(P_j) \right). \tag{11}$$

Since $P_j$ is not a vector with all ones, the spectral radius of $\tilde{A}_j$ is smaller than that of $\tilde{A}$, i.e.,

$$\rho(\tilde{A}_j) < \rho(\tilde{A}) = 1, \tag{12}$$

where the spectral radius is denoted by $\rho(\cdot)$. As the number of layers increases, node features do not collapse to the dominant eigenvector direction, and instead retain contributions from multiple eigenvector components. From the perspective of information propagation, this design effectively limits the extent of feature diffusion, allowing the model to focus on specific local regions with higher attention weights, thus mitigating uncontrolled propagation and preserving distinct local node information. Information propagation through low-attention nodes is significantly suppressed by the attention weights, limiting its influence. While deeper layers may reduce feature magnitudes, this scaling issue can be addressed by normalization methods such as LayerNorm. Moreover, distinct attention vectors $P_j$ in each view guide the model to focus on different local regions. Fusing these diverse perspectives further enhances the expressiveness and robustness of node representations.

Moreover, unlike global readout functions and traditional matrix decomposition or low-rank approximation methods, the alignment-based readout mechanism does not simply compress node features. Instead, it adaptively establishes correspondences between node clusters, enabling the use of a more expressive readout function such as an MLP. This design preserves permutation invariance while maintaining distinct structural information, thereby enhancing the discriminative power of graph representations and effectively mitigating the graph-level over-smoothing problem.

Here we select two representative toy examples of molecular graphs, ethanol and dimethyl ether, to compare the conventional GCNs with our proposed MultiNet model (see Figure 3). As shown on the left, in the traditional GCNs, multi-layer message passing causes node features to become homogenized, resulting in the loss of crucial local-level information. Subsequent readout operations (e.g., sum or mean pooling) further overlook structural differences between molecules, ultimately rendering the two distinct molecular graphs indistinguishable. In contrast, as shown on the right, our MultiNet leverages a subgraph convolution module that adaptively identifies and focuses on discriminative functional groups, guiding the convolutional process toward structurally distinct local regions and effectively reducing the similarity among node features. Furthermore, by incorporating a structure-aware node alignment strategy and employing a more expressive MLP, our method enhances the discriminative power of the final graph representations.

## 5 Experiments

### 5.1 The Experimental Setup

We evaluate the proposed MultiNet model on several benchmark graph classification datasets, including MUTAG, PTC_MR, ENZYMES, PROTEINS, DD, IMDB-B, and IMDB-M. These datasets

Table 1: Classification accuracy (In % ± standard error) on benchmark datasets.

| Method | MUTAG | PTC_MR | ENZYMES | PROTEINS | DD | IMDB-B | IMDB-M | Rank |
|---|---|---|---|---|---|---|---|---|
| DGCNN | 84.0±6.7 | 58.3±7.0 | 38.9±5.7 | 72.9±3.5 | 76.6±4.3 | 69.2±3.0 | 45.6±3.4 | 12.40 |
| DiffPool | 79.8±7.1 | 60.8±7.0 | 59.5±5.6 | 73.7±3.5 | 75.0±3.5 | 68.4±3.3 | 45.6±3.4 | 10.80 |
| ECC | 75.4±6.2 | 55.7±3.3 | 29.5±8.2 | 72.3±3.4 | 72.6±4.1 | 67.7±2.8 | 43.5±3.1 | 15.60 |
| GIN | 84.7±6.7 | 58.8±5.5 | **59.6±4.5** | 73.3±4.0 | 75.3±2.9 | 71.2±3.9 | 48.5±3.3 | 7.40 |
| GraphSAGE | 83.6±9.6 | 60.1±4.7 | 58.2±6.0 | 73.0±4.5 | 72.9±2.0 | 68.8±4.5 | 47.6±3.5 | 10.80 |
| DGK | 82.66±1.45 | 57.32±1.13 | 53.4±0.9 | 71.68±0.50 | 78.50±0.22 | 66.96±0.56 | 44.55±0.52 | 10.57 |
| 1-RWNN | 89.2±4.3 | — | 56.7±5.2 | 74.7±3.3 | 77.6±4.7 | 70.8±4.8 | 47.8±3.8 | 6.33 |
| 2-RWNN | 88.1±4.8 | — | 57.4±4.9 | 74.1±2.8 | 76.9±4.6 | 70.6±4.4 | 48.8±2.9 | 7.16 |
| 3-RWNN | 88.6±4.1 | — | 57.6±6.3 | 74.3±3.3 | 77.4±4.9 | 70.7±3.9 | 47.8±3.5 | 6.50 |
| GKNN-WL | 85.73±2.70 | 59.29±2.54 | — | 74.94±1.10 | — | 69.70±2.20 | 47.87±1.78 | 7.00 |
| GKNN-GL | 85.24±2.28 | 60.13 ±1.94 | — | 75.36±1.12 | — | 69.90±2.20 | 45.67±1.22 | 7.40 |
| RWGK | 80.77±0.72 | 55.91±0.37 | 22.37±0.35 | 74.20±0.40 | 71.70±0.47 | 67.94±0.77 | 46.72±0.30 | 13.14 |
| SPGK | 83.38±0.31 | 56.55±0.53 | 29.00±0.48 | 75.10±0.50 | 78.45±0.26 | 71.26±1.04 | 51.33±0.57 | 6.71 |
| GK | 81.66±0.11 | — | 24.87±0.22 | 71.67±0.55 | 78.45±0.26 | 65.87±0.98 | 45.42±0.87 | 14.17 |
| WLSK | 82.88±0.57 | 56.05±0.51 | 52.75±0.44 | 73.52±0.43 | **79.78±0.36** | 71.88±0.77 | 49.50±0.49 | 7.14 |
| JTQK | 85.50±0.55 | 57.39±0.46 | 56.41±0.42 | 72.86±0.41 | 79.49±0.32 | 72.45±0.81 | 50.33±0.49 | 6.00 |
| ASK | 87.50±0.65 | — | — | — | 70.38±0.22 | — | 50.12±0.51 | 9.33 |
| EDBMK | 86.35 | 56.75 | 36.85 | — | 78.19 | — | — | 8.25 |
| QBMK | 88.55±0.43 | 59.38±0.36 | — | — | 77.60±0.47 | — | — | 5.00 |
| **MultiNet** | **89.81±1.46** | **62.65±0.88** | 54.83±1.55 | **76.40±0.87** | 78.90±0.51 | **76.49±0.60** | **51.93±0.25** | **2.28** |

cover two primary domains: bioinformatics (Bio) and social networks (SN). All experiments are implemented in PyTorch and PyTorch Geometric, and executed on an NVIDIA GeForce RTX 3090 GPU (24GB VRAM). For all datasets, we use a local subgraph convolution module with 3 to 4 layers, set the number of views to 8, the node embedding dimension to 32, and the number of aligned nodes during readout to 8, with ReLU as the activation function. The model is trained using the Adam optimizer, with hyperparameters such as learning rate and number of epochs tuned via validation. To ensure statistical robustness, we perform ten runs of 10-fold cross-validation and report the mean classification accuracy along with the standard deviation [2].

## 5.2 Experiments on Graph Classification

We compare the MultiNet with several advanced GNNs and graph kernels. Specifically, **The advanced GNNs include**: five baseline models (the DGCNN [39], DiffPool [38], ECC [30], GIN [34], and GraphSAGE [16]) as well as six additional advanced models (the DGK [37], p-RWNN [23] (with $p = 1, 2, 3$), GKNN-WL, and GKNN-GL [11]). **The graph kernels include**: the RWGK [15], SPGK [6], GK [29], WLSK [28], JTQK [4] (where $q = 2$), ASK [5], EDBMK [35], and QBMK [3]. The classification accuracy and standard error are reported in Table 1, and the last column represents the average rank. For the baseline deep learning methods, we report the results from the fair comparison [14] or reproduce them following its evaluation protocol. For other models, we adapt their best results from their original papers.

The MultiNet achieves the best or near-best classification accuracy on most benchmark datasets, demonstrating the effectiveness of the proposed multi-view adaptive propagation mechanism in enhancing the performance of graph classification. The advantages of the MultiNet are summarized as follows: 1) **Compared with the graph kernel methods**, the MultiNet provides an end-to-end framework that adaptively extracts features from different views, eliminating the need for handcrafted kernel design and feature engineering. 2) **Compared to the traditional GNN-based methods**, our MultiNet restricts the extent of message propagation through node-level attention weights, enabling the model to focus on view-specific critical local features. This effectively suppresses redundant neighbor information and reduces noise. Moreover, MultiNet maintains multiple parallel subgraph convolutions, allowing information to propagate adaptively in different directions. This design captures diverse structural patterns and improves generalization across various types of graph data. 3) **Compared to the models that use simple readout functions (e.g., the GIN)**, the MultiNet introduces an alignment-based readout strategy that considers local node features and the overall graph structure when generating graph representations, thereby further enhancing the expressiveness.

---

[2]Code is available at https://github.com/Xiaoqin0421/MultiNet

Table 2: AD of initial node features for different datasets.

| Dataset | MUTAG | PTC_MR | ENZYMES | PROTEINS | DD | IMDB-B | IMDB-M |
|---------|-------|--------|---------|----------|-----|--------|--------|
| AD | 0.0331 | 0.1608 | 0.1315 | 0.1857 | 0.1437 | 0.6320 | 0.7328 |

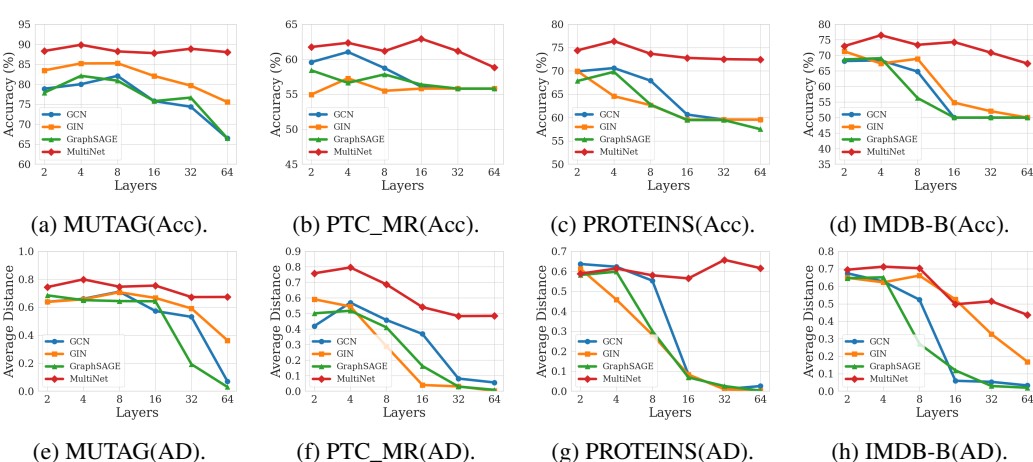

(a) MUTAG(Acc).  (b) PTC_MR(Acc).  (c) PROTEINS(Acc).  (d) IMDB-B(Acc).

(e) MUTAG(AD).  (f) PTC_MR(AD).  (g) PROTEINS(AD).  (h) IMDB-B(AD).

Figure 4: Classification accuracies (%) and the AD values on the four datasets.

## 5.3 Evaluations on Mitigating the Over-Smoothing

We conduct experiments to validate the effectiveness of MultiNet in mitigating the graph-level over-smoothing. To quantify the differences between graph representations, we adopt the Average Cosine Distance (AD). Given the graph-level representation matrix $H$, AD is defined as:

$$\mu(H) = \frac{1}{N^2} \sum_{i=1}^{N} \sum_{j=1}^{N} \left( 1 - \frac{H_i^\top H_j}{\|H_i\| \|H_j\|} \right). \tag{13}$$

Although this metric does not satisfy conditions a) and b) in Definition 1, our primary concern is whether graph representations converge to similar directions. Therefore, we choose a metric that emphasizes directionality rather than vector magnitudes, which is similar to those used in the node-level over-smoothing analysis. A higher AD value indicates greater diversity among graph representations, suggesting that over-smoothing is less severe. To analyze the impact of initial features, we compute the average node features for each dataset and calculate their corresponding AD values (see Table 2). The results show that the AD values for the Bio datasets are below 0.2, supporting Assumption 1 that the initial node features in these datasets are highly similar.

Subsequently, we compare the proposed MultiNet with three standard GCNs (including the Vanilla GCN, GIN, and GraphSAGE) on three Bio datasets (including the MUTAG, PTC_MR, and PRO-TEINS datasets) and an SN dataset: the IMDB-B dataset. We set the number of convolutional layers to 2, 4, 8, 16, 32, and 64, and report both classification accuracy (%) and the AD values in Figure 4. In the MultiNet, the number of layers refers to the depth of the local subgraph convolution module. The results show that our model already outperforms other baselines under shallow configurations (e.g., $L = 2 \sim 4$), demonstrating that the proposed subgraph convolution and the alignment-based readout are inherently effective without requiring deeper networks. As the network depth increases, the MultiNet experiences a smaller drop in accuracy and consistently maintains a higher AD value. In contrast, the AD values of the baseline models decrease significantly, indicating a more severe over-smoothing problem. These findings demonstrate that the MultiNet effectively mitigates the over-smoothing problem on the graph level. Even though Assumption 1 does not hold clearly on the IMDB-B dataset, the graph-level over-smoothing remains evident. This suggests that local node-level features and structural information play a crucial role in graph classification, which are effectively captured by the MultiNet. Notably, under deeper configurations (e.g., L=16 or L=32), the MultiNet achieves competitive or even superior classification accuracy compared to its shallower variants on certain datasets, which further demonstrates its ability to mitigate over-smoothing while preserving the advantages of deeper propagation.

## 5.4 Ablation Studies

We conduct ablation studies by removing individual modules to evaluate the contribution of each component in the proposed MultiNet. Specifically, we compare the performance of the following three variants: **(1) w/o L**: removing the local subgraph convolution module; **(2) w/o F**: replacing the feature fusion module with simple feature addition; **(3) Avg. Readout**: replacing the alignment module with an average readout function; and **(4) Sum Readout**: replacing the alignment module with a sum readout function. The results in Table 3 indicate that the MultiNet consistently outperforms all ablated variants across benchmark datasets. These findings demonstrate the effectiveness of each component in our framework. The local subgraph convolution module enables adaptive multi-view feature extraction. The feature fusion module effectively integrates information from diverse perspectives. Meanwhile, the alignment module enhances the expressiveness of the graph-level representations. Together, these modules collectively contribute to the superior classification performance.

Table 3: Ablation experimental accuracy (In % $\pm$ standard error).

| Method | MUTAG | PTC_MR | ENZYMES | PROTEINS | DD | IMDB-B | IMDB-M |
|---|---|---|---|---|---|---|---|
| MultiNet | **89.81±1.46** | **62.65±0.88** | **54.83±1.55** | **76.40±0.87** | **78.90±0.51** | **76.49±0.60** | **51.93±0.25** |
| w/o L | 83.39±1.04 | 58.83±0.54 | 27.33±1.05 | 75.01±0.40 | 78.60±0.45 | 74.73±0.61 | 50.61±0.43 |
| w/o F | 88.13±2.13 | 59.51±2.19 | 50.17±1.20 | 74.83±0.90 | 78.58±0.36 | 72.20±1.06 | 50.80±1.45 |
| Avg. Readout | 86.11±1.97 | 61.79±1.51 | 52.33±2.09 | 74.83±0.42 | 78.63±0.25 | 73.52±1.05 | 50.13±0.32 |
| Sum Readout | 85.55±0.96 | 62.53±0.58 | 29.17±1.77 | 73.51±1.42 | 78.42±0.32 | 73.50±1.92 | 47.13±1.43 |

## 5.5 Impact of the Number of Subgraphs

We evaluate the impact of the number of subgraphs on model performance by conducting extensive experiments on the MUTAG, PROTEINS, and IMDB-B datasets, varying the number of subgraphs among 2, 4, 8, 16, and 32. The results in Figure 5 show that setting the number to 8 yields the highest accuracy across all three datasets, suggesting that a moderate division effectively captures rich structural information and consistently enhances performance. However, further increasing the number of subgraphs (to 16 and 32) leads to performance degradation, possibly due to the introduction of noise or overfitting from excessive division.

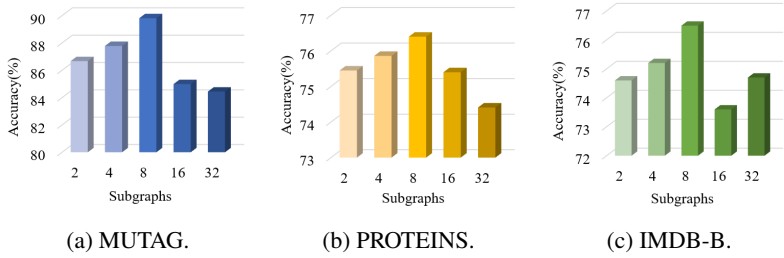

(a) MUTAG.      (b) PROTEINS.      (c) IMDB-B.

Figure 5: The experimental results of different numbers of subgraphs.

## 6 Conclusions

In this paper, we propose a novel MultiNet to mitigate the graph-level over-smoothing issue encountered in graph classification tasks. By incorporating a local subgraph convolution module, the MultiNet adaptively divides the input graph into multiple subgraph views. The proposed method extracts more discriminative local features by focusing on view-specific regions with higher attention weights, followed by cross-view node feature fusion to obtain comprehensive node representations. Additionally, we introduce an alignment-based readout mechanism that enhances the quality of the graph representations. Through theoretical analysis and experimental results, we demonstrate that the MultiNet effectively mitigates the graph-level over-smoothing and consistently outperforms existing state-of-the-art methods on graph classification tasks. In future work, we aim to extend our model to explore the over-smoothing problem in node classification tasks and investigate sparsification-based approaches to further reduce its spatial complexity.

## Acknowledgements

This work is supported by the National Natural Science Foundation of China (Nos. 62576371, T2122020, 62172370 and 62576198). Ming Li acknowledged the supports from the Jinhua Science and Technology Plan (No. 2023-3-003a). This work is also supported by Open Project Foundation of Key Laboratory of Computation Intelligence and Chinese Information Processing of Ministry of Education and Key Laboratory of Data Intelligence and Cognitive Computing of Shanxi Province.

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

# A Technical Appendices and Supplementary Material

## A.1 Statistics of the Datasets

In Table 4, we summarize the statistics of the benchmark graph classification datasets used in this work, including the maximum (Max #Nodes) and average number of nodes (Mean #Nodes), total number of graphs (#Graphs), number of classes (#Classes), and the domain of each dataset.

Table 4: Statistics of the datasets used in our experiments

| Dataset | MUTAG | PTC_MR | ENZYMES | PROTEINS | DD | IMDB-B | IMDB-M |
|---|---|---|---|---|---|---|---|
| Max #Nodes | 28 | 64 | 126 | 620 | 5748 | 136 | 89 |
| Mean #Nodes | 17.93 | 14.29 | 32.63 | 39.06 | 284.3 | 19.77 | 13 |
| #Graphs | 188 | 344 | 600 | 1113 | 1178 | 1000 | 1500 |
| #Classes | 2 | 2 | 6 | 2 | 2 | 2 | 3 |
| Domain | Bio | Bio | Bio | Bio | Bio | SN | SN |

## A.2 Visualization

To intuitively demonstrate the effectiveness of multi-view subgraph division, we showcase examples from real-world datasets in Figure 6, including MUTAG, PROTEINS, and IMDB-B. Each row corresponds to the same graph instance, while each column shows a specific subgraph view. The intensity of the node color represents the attention weights. The variation in color across different subgraphs suggests that the model adaptively regulates information propagation between subgraphs by capturing features from distinct local regions. This helps preserve local information and mitigates the over-smoothing of node features.

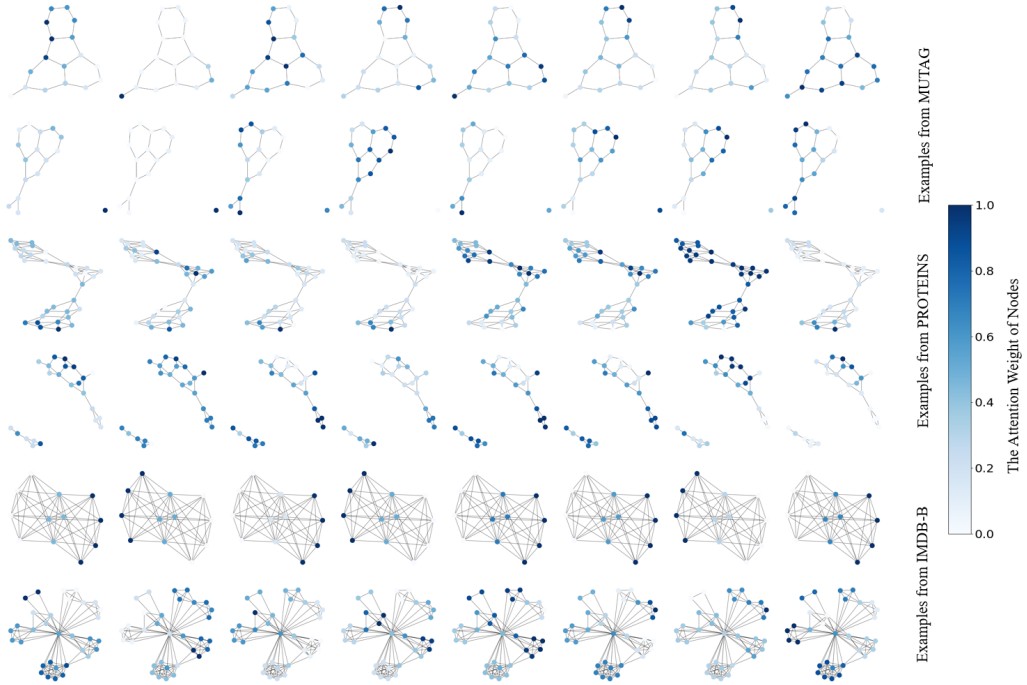

Figure 6: Examples of multi-view subgraph division from different datasets.

