# OpenReview forum: "MultiNet: Adaptive Multi-Viewed Subgraph Convolutional Networks for Graph Classification"
_NeurIPS.cc/2025/Conference — NeurIPS 2025 poster_

### Official Review · Reviewer_7WyT · 2025-06-30

**Clarity:** 4
**Significance:** 3
**Originality:** 4
**Rating:** 5
**Confidence:** 4

**Summary:**

This paper defines a new graph neural network, namely the Adaptive Multi-viewed Subgraph Convolutional Network (MultiNet), for graph classification. Unlike the classical GNNs, the new MultiNet is defined based on the subgraph convolution operation within the substructures. The substructures are divided by assign different attentions on the nodes, and each substructure consists of nodes with higher attention. Therefore, performing the convolution operation on the substructure essentially equals to performing a subgraph convolution, and this strategy can help to reduce the over-smoothing problem. Because, the information passing is limited among different subgraphs. The new MultiNet have better graph classification performance than the classical GNN methods.

**Questions:**

I think the authors need to answer my following questions, and the paper can be further improved.

First, Eq. 4 and Eq. 8 have the same form, what is the difference between them?

Second, the node-level over-smoothing is always discussed by many works, what is the difference between the node-level and graph-level over-smoothing?

Third, some minor grammar mistakes need to be fixed, such the sentence on row 184, “where the extent to which a node transmits its information to neighbors...”, there are two leading words of subordinate clause (where and which) .

Fourth, as I remember the MUTAG and PTC data was well evaluated by many classical methods, including the DGCNN and GIN, why the authors did not include the results？

**Ethical Concerns:**

["NO or VERY MINOR ethics concerns only"]

**Final Justification:**

Thank the authors for addressing my concerns. All of my questions have now been resolved. As I mentioned before, this work is valuable, and I will maintain my positive score.

**Limitations:**

I have listed some limitations in Questions, such as some grammar and statement problems.

**Paper Formatting Concerns:**

I do not have any concerns for the format.

**Quality:**

3

**Strengths And Weaknesses:**

Strengths:
An interesting idea to reduce the over-smoothing for graph classification.
Good idea to define new subgraph convolution operation with the attention mechanism.
Overall, this paper is clearly written and easy to follow.

Excluding the strengths, I still have some major concerns. Please see Questions.

---

> ### Author Rebuttal · Authors · 2025-07-30
>
> Dear Reviewer 7WyT,
>
> We sincerely thank you for your careful reading of our paper and for the valuable questions and suggestions. Below, we provide detailed responses to each of your comments. We have made every effort to clarify our contributions and address your concerns with additional explanations, experiments, or theoretical analysis where appropriate.
>
> We hope that our responses can resolve your concerns and better convey the significance of our work. We would greatly appreciate any additional feedback you may have.
>
> Response to Q1: Thanks for the suggestion. Although Eq. (4) and Eq. (8) appear similar in form, they serve entirely different purposes and carry distinct semantic meanings within our model. The attention matrix P in Eq. (4) is used during the multi-view division stage, where the nodes of the original graph are assigned to m subgraph views based on their attention scores. This process enables diverse local substructure modeling and emphasizes the relative importance of nodes across different perspectives. In contrast, the matrix S in Eq. (8) is introduced at the final representation generation stage as an alignment-based aggregation matrix. It maps the final node embeddings into s globally shared clusters to facilitate structural alignment and produce the graph-level representation. These two matrices correspond to two distinct phases in our architecture. While they may share a similar mathematical form, their usage and semantic roles are fundamentally different. We will clarify this distinction further in the revised version.
>
> Response to Q2: Thanks for the suggestion. Although the node-level and graph-level over-smoothing are closely related, they differ in their focus. The node-level over-smoothing occurs within a single graph, where repeated graph convolutions lead to overly similar node representations, ultimately degrading the ability to distinguish nodes for node classification tasks. In contrast, the graph-level over-smoothing refers to the phenomenon where the graph-level representations of different graphs become excessively similar after the readout stage, making it difficult for the model to distinguish between graphs and thus harming performance in graph classification. In our work, we have explicitly distinguished these two types of over-smoothing in Section 2, provided a formal definition and theoretical analysis of graph-level over-smoothing in Section 2.2, and offered an illustrative example in Figure 3. We will further enhance the discussion in the revised version.
>
> Response to Q3: Thanks for the suggestion. We will thoroughly review the entire manuscript to correct any grammatical errors and typographical mistakes.
>
> Response to Q4: Thanks for the suggestion. We have clearly indicated the source of the experimental results in Section 5.2. For the baseline deep learning methods, we follow the fair comparison protocol provided in [1], which does not report results on the MUTAG and PTC_MR datasets. Accordingly, we did not include these baselines on the two datasets in our initial submission. To address your concern, we have reproduced the results for these models using the official code from [1]. As shown in the table below, our proposed MultiNet still achieves superior performance on both datasets, further demonstrating its effectiveness and robustness. We will update the manuscript to incorporate additional results and relevant discussions.
>
> | Datasets | DGCNN | DiffPool| ECC | GIN| GraphSAGE| MultiNet|
> |----------|------------|------------|-----------|-----------|-----------|--------------|
> | MUTAG| 84.0±6.7| 79.8±7.1| 75.4±6.2| 84.7±6.7| 83.6±9.6| **89.81±1.46** |
> | PTC_MR| 58.3±7.0| 60.8±7.0| 55.7±3.3| 58.8±5.5| 60.1±4.7| **62.65±0.88** |
>
> [1] Federico Errica, Marco Podda, Davide Bacciu, and Alessio Micheli. A Fair Comparison of Graph Neural Networks for Graph Classification. In Proceedings of ICLR, 2020.

---

> > ### Comment · Reviewer_7WyT · 2025-08-04
> >
> > Thank you to the authors for addressing my concerns. All of my questions have now been resolved. As I mentioned before, this work is valuable, and I will maintain my positive score.

---

> > > ### Author Response · Authors · 2025-08-05
> > >
> > > We appreciate the reviewer's efforts in reading our responses. We will revise the paper based on the constructive suggestions for the final version.

---

### Official Review · Reviewer_YyPC · 2025-07-01

**Clarity:** 3
**Significance:** 3
**Originality:** 3
**Rating:** 4
**Confidence:** 4

**Summary:**

This paper introduces MultiNet, a novel Adaptive Multi-Viewed Subgraph Convolutional Network designed to address graph-level over-smoothing in Graph Convolutional Networks (GCNs). By leveraging local subgraph convolution and alignment-based readout mechanisms, MultiNet enhances graph classification performance through preserving local node features and structural information.

**Questions:**

Q1. How does the paper theoretically or experimentally prove that permutation invariance of traditional readout functions is the primary cause of graph-level over-smoothing?
Q2. Given that the proposed model still relies on permutation-invariant aggregation, how does it fundamentally address graph-level over-smoothing?

Q3. Does the proposed model leverage deeper propagation to achieve better performance, or does it primarily focus on mitigating performance degradation caused by deeper layers?

Q4. Are there any experiments showing that the model outperforms shallower networks in terms of classification accuracy?

Q5. Why does the paper not include comparisons with state-of-the-art graph-level algorithms? How does the proposed method optimize graph-level representations compared to these recent approaches?

Q6. How does the aggregation process in the proposed model relate to matrix factorization techniques? Can the authors provide theoretical or experimental validation to demonstrate the effectiveness of using matrix factorization-like operations in the model design?

**Ethical Concerns:**

["NO or VERY MINOR ethics concerns only"]

**Final Justification:**

This rebuttal partially resolved my concerns. Despite my consistent questioning about the permutation-invariant aggregation, I would like to raise my score due to the reviews given by the other reviewers. I will also not be against the acceptance of this paper.

**Limitations:**

Yes.

**Quality:**

3

**Strengths And Weaknesses:**

S1. MultiNet adaptively divides graphs into multiple subgraph views, capturing discriminative local node features and mitigating node-level over-smoothing.

S2. Establishes node correspondences across graphs, enabling expressive readout functions that preserve structural information and reduce graph-level over-smoothing.

S3. Demonstrates effectiveness in mitigating over-smoothing and achieving state-of-the-art graph classification performance on benchmark datasets.

W1. The paper claims that solving node-level over-smoothing cannot fully address graph-level over-smoothing due to the permutation invariance of traditional readout functions. However, the authors do not provide sufficient theoretical or experimental evidence to support this claim. Moreover, the proposed model still relies on permutation-invariant aggregation, which raises questions about whether the issue is truly resolved.

W2. The ultimate goal of addressing over-smoothing is to improve model performance. However, the experimental results show that the proposed model primarily mitigates the performance degradation caused by deeper layers, rather than leveraging deeper propagation to achieve better overall performance. This limits the practical benefits of the model.

W3. The paper lacks comparisons with recent graph-level algorithms. Without such comparisons, it is difficult to evaluate the effectiveness of the proposed method in optimizing graph-level representations.

W4. The aggregation process in the proposed model closely resembles matrix factorization, particularly in the use of the node assignment matrix to map node embeddings to cluster-level embeddings. However, the paper does not explicitly discuss this similarity or provide theoretical validation for the effectiveness of this design. Adding a discussion and validation of matrix factorization approximations would strengthen the paper's contributions.

---

> ### Author Rebuttal · Authors · 2025-07-30
>
> Dear Reviewer YyPC,
>
> We sincerely thank you for your careful reading of our paper and for the valuable questions and suggestions. Below, we provide detailed responses to each of your comments. We have made every effort to clarify our contributions and address your concerns with additional explanations, experiments, or theoretical analysis where appropriate.
>
> We hope that our responses can resolve your concerns and better convey the significance of our work. We would greatly appreciate any additional feedback you may have.
>
> Response to Q1: Thanks for the suggestion. We argue that our focus is not on permutation-invariant readout functions causing graph-level over-smoothing, but rather on how overly simplistic global readout functions (e.g., sum or average pooling) can exacerbate the graph-level over-smoothing when the node representations become homogeneous. Specifically, graph classification tasks inherently require the permutation-invariant readouts to ensure the output consistency regardless of the node ordering. However, most traditional GNNs often rely on the overly simplified global pooling schemes that treat all nodes as equally important. This approach ignores both the local node features and the topological structures. When combined with the node-level over-smoothing, it results in indistinguishable graph representations and severely degrades the discriminative power of the model. To support this claim, we have provided a formal mathematical proof in Theorem 1 (Section 2.2), which shows how the simple readout functions can lead to the graph-level representation convergence under the node feature homogenization. In addition, we offer an illustrative example in Figure 3 (left) and empirical evidence in Section 5.4 through the ablation studies comparing alignment readout with the traditional avg/sum pooling. We will revise the relevant discussion in the final version to ensure clarity and rigor.
>
> Response to Q2: Thanks for the suggestion. We agree that the permutation invariance is a necessary property for the readout functions in graph classification, as the predicted graph labels should not depend on the node ordering. However, we argue that the issue lies not in the permutation invariance itself, but in the simplicity of the traditional permutation-invariant global readout functions, such as sum or average pooling, which treat all nodes equally. The root reason of this issue lies in the disorder of nodes within the graph. To address this, we aim to establish a reliable ordered correspondence for node clusters, which leads us to introduce an alignment-based readout mechanism. Specifically, we learn a cluster assignment matrix S using a shared function across all graphs, which ensures that clusters are aligned in a consistent order across different samples. This aligned structure allows us to apply a non-symmetric MLP at the cluster level, assigning different importance weights to different clusters. This mechanism preserves permutation invariance at the graph level, while avoiding the uniform treatment of nodes, thereby improving the discriminative power of the graph representation and effectively mitigating the graph-level over-smoothing. For this issue, we have clearly explained our motivation in the Introduction and provided a theoretical explanation in Section 4. We will further elaborate on this mechanism in the revised version.
>
> Response to Q3: Thanks for the suggestion. Our goal is to enhance the discriminative power of graph representations by alleviating the graph-level over-smoothing, rather than relying solely on the deeper propagation for performance gains. As shown in Table 1 and Figure 4, our model already outperforms other baselines under shallow configurations (e.g., L=2~4), demonstrating that the proposed subgraph convolution and the alignment-based readout are inherently effective without requiring deeper networks. Furthermore, the trends in Figure 4 indicate that our model degrades much more slowly as the number of layers increases, effectively alleviating the performance deterioration typically observed in deeper GCNs. Notably, in some datasets, deeper versions (e.g., L=16 or L=32) even outperform shallower ones. This shows that our method not only improves the stability in deeper architectures but also retains the potential to benefit from deeper propagation. Therefore, we believe our model is practically valuable, as it adapts well to both shallow and deep settings, offering robust and competitive graph-level representations. We will follow your suggestion and revise the relevant sections to enhance clarity.
>
> Response to Q4: Thanks for the suggestion. In Figure 4, we conduct a systematic comparison across different network depths. The results show that our method consistently achieves higher classification accuracy than other models under both shallow and deep settings. Notably, under deeper configurations (e.g., L=16 or L=32), our MultiNet achieves competitive or even superior classification accuracy compared to its shallower variants on certain datasets, which further demonstrates its ability to mitigate over-smoothing while preserving the advantages of deeper propagation. We will follow your suggestion and revise the relevant sections to enhance clarity.
>
> Response to Q5: Thanks for the suggestion. In Table 1 of our paper, we have included over eighteen representative and state-of-the-art graph classification models, such as p-RWNN (2020), GKNN-WL (2024), GKNN-GL (2024), and QBMK (2024), covering both recent graph neural networks and graph kernel approaches. The results demonstrate that our proposed MultiNet consistently achieves superior classification accuracy across various datasets. Furthermore, Section 5.2 provides a comparative analysis of graph-level representation effectiveness, while the ablation studies in Section 5.4 confirm that each component of our model plays a crucial role in enhancing representation quality. We appreciate the suggestion and will include additional comparisons with more recent methods in the revised version to further strengthen the empirical validation.
>
> Response to Q6: Thanks for the suggestion. Our method is fundamentally different from the traditional matrix factorization in both its motivation and computational process. First, the core objective of our alignment-based readout is to establish the consistent cluster-level correspondences across graphs, thereby overcoming the limitations of the conventional global readout functions in preserving structural information. This design enhances the expressiveness of the graph-level representations by explicitly retaining the local features and structural distinctions, rather than merely compressing node embeddings. Therefore, the process cannot be simply interpreted as the feature compression or the low-rank approximation. Second, in terms of the computational mechanism, the node-to-cluster assignment matrix is adaptively learned in an end-to-end fashion based on both the graph structure and node features. This assignment is not a static matrix nor derived from a predefined decomposition objective. Consequently, we consider our design conceptually distinct from classical matrix factorization techniques. We have already discussed and justified this design choice in the Introduction and further elaborated it in Section 3.4. We will include a more detailed discussion of this distinction in the revised version to improve theoretical clarity.

---

> ### Comment · Reviewer_YyPC · 2025-08-04
>
> I would like to thank the authors for the response.
> This rebuttal partially resolved my concerns. Despite my consistent questioning about the permutation-invariant aggregation, I would like to raise my score due to the reviews given by the other reviewers. I will also not be against the acceptance of this paper.

---

> > ### Comment · Reviewer_YyPC · 2025-08-04
> >
> > After carefully reading some other related works, including GKNN-WL (2024), GKNN-GL (2024), and QBMK (2024) mentioned in the rebuttal, I believe my focus differs significantly from the authors'. I would like to thank the authors for their careful rebuttal. Now I would like to support the acceptance of this paper.

---

> > > ### Author Response · Authors · 2025-08-04
> > >
> > > We appreciate the reviewer's efforts in reading our responses. We will revise the paper based on the constructive suggestions for the final version.

---

### Official Review · Reviewer_xLjw · 2025-07-01

**Clarity:** 3
**Significance:** 3
**Originality:** 4
**Rating:** 5
**Confidence:** 5

**Summary:**

This paper develops a new Adaptive Multi-Viewed Subgraph Convolutional Network (MultiNet) to solve the over-smoothing problem and improve the graph classification performance. The MultiNet is defined based on the subgraph convolution on divided subgraphs from the original global graph. The experiments on standard datasets confirm the performance of the MultiNet. Overall, this paper is well-written, easy to follow and  and technically sound.

**Questions:**

Some statements need to be more clear. Some experiments are missed. Some technology details need to be well explained. Please refer my rebuttal weakness for details.

**Ethical Concerns:**

["NO or VERY MINOR ethics concerns only"]

**Final Justification:**

All my concerns have been well addressed. This work is theoretically solid and proposes a novel method. Therefore, I recommend accepting this paper.

**Limitations:**

Yes.

**Paper Formatting Concerns:**

I didn’t see any obvious formatting problem.

**Quality:**

3

**Strengths And Weaknesses:**

**Strengths**
1. This paper defines an end-to-end framework to adaptively divide the input graph into subgraph views, and perform the local convolution on the subgraphs to limit the information passing among different subgraph views. So, the new MultiNet can avoid the over-smoothing problem.
2. This paper defines a new local convolution filter on the subgraph view with the attentions of adjacent nodes. The new filter can propagate the node information based on the attention and limit the information of nodes with lower attention.
3. The authors provide necessary theoretical analysis to confirm the advantages of  the new MultiNet.

**Weaknesses**
1. The over-smoothing is always a serious problem for node classification, but how the over-smoothing influence the graph classification? More theoretical analysis or descriptions are needed.
2. Why use the alignment readout, why do not directly use the sum or avg pooing? These operations are widely used for graph neural networks to extract graph representation.
3. For table 1, the accuracies of some method are missed, why not compare them?
4. The authors also need to cite some necessary references about graph over-smoothing problem, not only list the references about node over-smoothing.

---

> ### Author Rebuttal · Authors · 2025-07-30
>
> Dear Reviewer xLjw,
>
> We sincerely thank you for your careful reading of our paper and for the valuable questions and suggestions. Below, we provide detailed responses to each of your comments. We have made every effort to clarify our contributions and address your concerns with additional explanations, experiments, or theoretical analysis where appropriate.
>
> We hope that our responses can resolve your concerns and better convey the significance of our work. We would greatly appreciate any additional feedback you may have.
>
> Response to W1: Thanks for the suggestion. Although over-smoothing was originally identified in the context of node classification, we argue that it also has a significant impact on graph classification. When stacking multiple graph convolution layers, node features across structurally different regions may become overly similar, thereby diminishing the ability to preserve and distinguish informative local substructures. This issue is further exacerbated by the simple permutation-invariant global readout functions (e.g., sum or mean), which can obscure the structural differences among graphs and reduce the discriminative power of graph-level representations. To address this, we provide a formal definition of graph-level over-smoothing in Section 2.2, along with a theoretical analysis of its causes and implications. In addition, Figure 3 offers an illustrative example showing how over-smoothing degrades the distinctiveness of graph representations in classification tasks. We will strengthen this theoretical discussion further in the revised version.
>
> Response to W2: Thanks for the suggestion. Traditional pooling operations such as sum, mean, or max are indeed widely used due to their simplicity, but they tend to overlook the local structural and the feature differences among nodes. In contrast, our alignment-based readout learns a cluster assignment matrix that aligns the order of node clusters and preserves the local topology across graphs, enabling the use of a more expressive MLP mapping. This enhances the ability to capture discriminative subgraph structures. We have highlighted these advantages in Section 3.4 and Section 4, and further validated the effectiveness of the alignment readout by replacing it with average and sum readouts in the ablation studies presented in Section 5.4. We will improve the explanation of this design choice in the revised version.
>
> Response to W3: Thanks for the suggestion. We have clearly indicated the source of the experimental results in Section 5.2. To ensure a fair comparison, we report the results from the fair comparison [1] for the baseline deep learning methods. For other models, we adapt their best results from their original papers. Since the referenced literature lacked comparative analyses on these specific datasets, we did not initially include their experimental results. Following the experimental protocol established in [1], we have preliminarily supplemented the performance evaluation of baseline deep learning methods on both MUTAG and PTC_MR datasets. As shown in the table below, our proposed MultiNet still achieves superior performance on both datasets, further demonstrating its effectiveness and robustness. We will update the manuscript to incorporate additional results and relevant discussions.
>
> | Datasets | DGCNN | DiffPool| ECC | GIN| GraphSAGE| MultiNet|
> |----------|------------|------------|-----------|-----------|-----------|--------------|
> | MUTAG| 84.0±6.7| 79.8±7.1| 75.4±6.2| 84.7±6.7| 83.6±9.6| **89.81±1.46** |
> | PTC_MR| 58.3±7.0| 60.8±7.0| 55.7±3.3| 58.8±5.5| 60.1±4.7| **62.65±0.88** |
>
> [1] Federico Errica, Marco Podda, Davide Bacciu, and Alessio Micheli. A Fair Comparison of Graph Neural Networks for Graph Classification. In Proceedings of ICLR, 2020.
>
> Response to W4: Thank you for the suggestion. Most existing literature has primarily focused on the node-level over-smoothing, and there is a notable scarcity of studies addressing the over-smoothing problem at the graph level. In our paper, we take a step forward by providing a strict definition of the graph-level over-smoothing for the first time in Section 2.2. Furthermore, in Section 5.3, we introduce a scientific metric to measure the graph-level over-smoothing and validate it through experiments.
> Although the work in [2] proposes a strategy to mitigate over-smoothing and applies it for graph classification tasks, it still focuses on alleviating over-smoothing at the node level and lacks the experiments addressing over-smoothing at the graph level. Thus, our work fills this gap and offers a more comprehensive analysis. We will provide additional explanations in the revised version to enhance our contribution.
>
> [2] Stevan Stanovica, Benoit Gaüzère, and Luc Brun. Graph Neural Networks with maximal independent set-based pooling: Mitigating over-smoothing and over-squashing. Pattern Recognit. Lett., 187, 14–20, 2025.

---

> > ### Comment · Reviewer_xLjw · 2025-08-08
> >
> > Thanks for the authors' detailed response. My concerns have been well addressed. I think this is a well-motivated and solid work. I support the acceptance of this paper.

---

> > > ### Author Response · Authors · 2025-08-09
> > >
> > > We appreciate the reviewer's efforts in reading our responses. We will revise the paper based on the constructive suggestions for the final version.

---

### Official Review · Reviewer_XtUo · 2025-07-01

**Clarity:** 3
**Significance:** 4
**Originality:** 3
**Rating:** 5
**Confidence:** 4

**Summary:**

This work aims to reduce the effects of the graph-level over-smoothing problem for graph classification, and proposes a novel MultiNet model (a new graph neural network, GNN). This paper first analyze how the over-smoothing influences the graph representation learning through the GNN. Then, this paper proposes to perform the attention-based subgraph convolution on adaptive extracted subgraph based views, that focuses on the information propagation within the subgraph view and naturally reduce the information passing over the global structure. The experiments on benchmark datasets show the superior performance for the proposed MultiNet model.

**Questions:**

1. I feel a little confused for the symbol P^(j) between the equations (11) and (12), should it be P_j ? It is not consistent to other symbol P_j. Maybe, it is a typos?
2. For the experiments, why some GNN models are not evaluated on the MUTAG and PTC datasets?

**Ethical Concerns:**

["NO or VERY MINOR ethics concerns only"]

**Final Justification:**

accept

**Limitations:**

I think some theoretical analysis about how the proposed method reduce the graph-level over-smoothing need to be more clear. As I say in the weakness, the information seems still pass between low attention nodes, this may lead to some minor over-smoothing problem.

**Quality:**

3

**Strengths And Weaknesses:**

Strengths:
1. An interesting strategy to divide the global graph structure into subgraph-based views, and perform the subgraph convolution to reduce the over-smoothing problem of the convolution operation on the global structure.
2. Provide mathematical analysis to show how the MultiNet works and limits the over-smoothing over the global structure.
 Weakness:
It seems that each subgraph view is not completely isolated from the global graph structure, since the substructure is identified by node attentions. Although the information may be mainly propagated between high attention nodes, the information may still pass between the low attention nodes. The authors did not clearly explain this problem.

---

> ### Author Rebuttal · Authors · 2025-07-30
>
> Dear Reviewer XtUo,
>
> We sincerely thank you for your careful reading of our paper and for the valuable questions and suggestions. Below, we provide detailed responses to each of your comments. We have made every effort to clarify our contributions and address your concerns with additional explanations, experiments, or theoretical analysis where appropriate.
>
> We hope that our responses can resolve your concerns and better convey the significance of our work. We would greatly appreciate any additional feedback you may have.
>
> Response to weakness and limitations: Thanks for the suggestion. In our design, we adopt a soft attention mechanism instead of the hard assignment to ensure differentiability and stable gradient backpropagation. The adaptive node attention allows the model to emphasize important nodes in each subgraph view, guiding it to focus on more discriminative local regions in terms of structure. While some information may still flow between the low-attention nodes, such propagation is effectively suppressed by the attention weights, which significantly limits its influence. This constrained message propagation mechanism mitigates the global feature homogenization and thereby alleviates the graph-level over-smoothing issue. We will further refine our theoretical analysis in section 4 to clearly explain how our method suppresses over-smoothing while preserving the flexible information propagation, as well as discussing the potential information flow among the low-attention nodes.
>
> Response to Q1: Thanks for the suggestion. The notation $P^(j)$ should be written as $P_j$, which denotes the attention vector for the j-th subgraph view. We will correct this notation in the final version.
>
> Response to Q2: Thanks for the suggestion. We have clearly indicated the source of the experimental results in Section 5.2. For the baseline deep learning methods, we follow the fair comparison protocol provided in [1], which does not report results on the MUTAG and PTC_MR datasets. Accordingly, we did not include these baselines on the two datasets in our initial submission. To address your concern, we have preliminarily reproduced the results for these models using the official code from [1]. As shown in the table below, our proposed MultiNet still achieves superior performance on both datasets, further demonstrating its effectiveness and robustness. We will update the manuscript to incorporate additional results and relevant discussions.
>
> | Datasets | DGCNN | DiffPool| ECC | GIN| GraphSAGE| MultiNet|
> |----------|------------|------------|-----------|-----------|-----------|--------------|
> | MUTAG| 84.0±6.7| 79.8±7.1| 75.4±6.2| 84.7±6.7| 83.6±9.6| **89.81±1.46** |
> | PTC_MR| 58.3±7.0| 60.8±7.0| 55.7±3.3| 58.8±5.5| 60.1±4.7| **62.65±0.88** |
>
> [1] Federico Errica, Marco Podda, Davide Bacciu, and Alessio Micheli. A Fair Comparison of Graph Neural Networks for Graph Classification. In Proceedings of ICLR, 2020.

---

> > ### Comment · Reviewer_XtUo · 2025-08-05
> >
> > The authors have addressed all of my concerns. I maintain my positive evaluation.

---

> ### Comment · Reviewer_XtUo · 2025-08-05
>
> Thanks for the response. I decide to keep the score unchanged.

---

> > ### Author Response · Authors · 2025-08-07
> >
> > We appreciate the reviewer's efforts in reading our responses. We will revise the paper based on the constructive suggestions for the final version.

---

### Decision · Program_Chairs · 2025-09-17

**Decision:**

Accept (poster)

**Comment:**

This paper proposes MultiNet, an adaptive multi-viewed subgraph convolutional network designed to mitigate graph-level over-smoothing and enhance graph classification. Reviewers initially raised questions regarding the distinction between node-level and graph-level over-smoothing, the clarity of theoretical analysis, missing baselines, and the design of the alignment-based readout. The authors’ rebuttal provided thorough clarifications, additional experiments, and theoretical explanations, which effectively addressed the reviewers’ concerns. Overall, I recommend acceptance, as the work is technically solid, though the final version would benefit from clearer exposition of the theoretical contributions and further emphasis on the differences from related matrix factorization approaches.